# A Projection of Future Hospitalisation Needs in a Rapidly Ageing Society: A Hong Kong Experience

**DOI:** 10.3390/ijerph16030473

**Published:** 2019-02-06

**Authors:** Xueyuan Wu, Chi-kin Law, Paul Siu Fai Yip

**Affiliations:** 1Department of Economics, The University of Melbourne, Parkville, VIC 3010, Australia; xueyuanw@unimelb.edu.au; 2Centre for Applied Health Economics, Menzies Health Institute Queensland, School of Medicine, Griffith University, Nathan, QLD 4111, Australia; c.law1@griffith.edu.au; 3Department of Social Work and Social Administration, The University of Hong Kong, Hong Kong, China

**Keywords:** Hospitalisation, ageing, Hong Kong

## Abstract

To assess the impact of ageing on hospitalisation in a rapidly ageing society. A study using retrospective and prospective data was conducted using hospitalisation data with age-specific admission rates in the period from 2001–2010 and demographic data from the period of 2001–2066 by the United Nations. The Hong Kong Special Administrative Region (SAR) with a 7 million population experiences extreme low fertility (1.1 children per woman) and long life expectancy (84 years old). Days of hospitalisation: For the period 2010–2066, the length of stay (LOS) in the age group 85+ is projected to increase by 555.3% while the LOS for the whole population is expected to increase by only 134.4% and by ageing only. In 2010, the proportion in the LOS contributed to by the oldest age group (85+) was 15%. In 2066, this proportion is projected to nearly triple (42%). Around 70% of the projected days of hospitalisation would be taken by people aged 75 years and above. It is projected that this phenomenon would be converted to a more balanced structure when the demographic transition changes into a more stable distribution. Apparently, the impact of ageing on the public hospital system has not been well understood and prepared. The determined result provides insight into monitoring the capacity of the hospital system to cope with a rapidly changing demographic society. It provides empirical evidence of the impact of ageing on the public hospitalisation system. It gives a long term projection up to the year 2066 while the situation would be different from the transient period of 2016–2030. The analysis adopts a fixed rate approach, which assumes the LOS to be only driven by demographic factors, while any improvements in health technologies and health awareness are not accounted for. Only inpatient data from the Hospital Authority were used, nonetheless, they are the best available for the study. Due to the limitation of data, proximity to death is not controlled in conducting this analysis.

## 1. Introduction

Due to prolonged below-replacement fertility and the prolongation of human lifespan, the unmanageable expansion in health care costs has become a major public finance issue in many countries [1,2]. Among different types of care, health care expenditures on hospital costs are always the most expensive, hence, many health care systems seek to reduce the patients’ duration of stay in hospitals in order to elevate the efficiency of the hospital system [3,4,5]. The increase in the number of older adults has had a great impact on hospital admissions [6,7,8,9]. More so in many high-income countries, population ageing is an inevitable phenomenon which certainly causes an increasing demand for medical and health care services. In the United States, about 17% of its gross domestic product (GDP) is spent on health care but health indicators are not particularly impressive, and life expectancy has been reducing since 2015 [10,11]. In the United Kingdom, the much acclaimed National Health System (NHS) is facing serious financial and sustainable problems, which ultimately lead to quality deterioration of hospital care services to patients [12]. 

Apparently, most of the present health care systems are unprepared for a rapidly ageing society. Patients’ profile and their duration of stay for each episode as well as the number of inpatient episodes per patient have been shown to be related to the total days of hospitalisation. One of the most effective methods in reducing the days of hospitalisation is to reduce potentially avoidable hospitalisation admissions and shorten the average length of stay (LOS) per episode through health technology advancements in addition to advocating new models of care in the primary care setting [3,5,13]. Nevertheless, how much further the LOS can be shortened without impacting the quality of care is difficult to assess. Furthermore, an increase in frail older adults would definitely increase the LOS. It is thus important to make better hospital planning for the expectant “tsunami” of older adults in the future.

As in many high-income Asian economies, due to ultra-low fertility and long life expectancy, Hong Kong faces an accelerated pace of population ageing. For the past three decades, there have been substantial changes in the demographic structure in Hong Kong. Due to the baby boom effect in the 1950–1960s and an influx of young immigrants from Mainland China during the 1970–1980s, from the period of 1986 to 2016, the total population of Hong Kong has grown by 33%, while those aged 65 years or above have tripled during the same period. This has led to the median age of the male population increasing from 28.7 years in 1986 to 43.7 years in 2016 and for the same period for females from 28.9 years to 43.2 years. In the next 56 years from 2010 to 2066, the population of Hong Kong will increase by 9.95% to 7.723 million, with 33.7% of the population aged 65 years or above [14]. The magnitude and speed of ageing in Hong Kong is unprecedented; it would take 100 years for France to double its older adult population whereas it would only take 25 years for Hong Kong. Rapid ageing is also not that uncommon especially among the high-earning in low fertility countries and regions in Asia, for example, in South Korea, Japan, and Taiwan. 

In addition, it is projected that for these economies, the ageing phenomenon would be worsened in the next 30 years under the extra-low fertility regime [15,16]. Given the status quo, demographics are well recognized as prominent driving forces in constituting the health needs of a population. Since 2000, the Hospital Authority (HA) fully funded by the Hong Kong Government, providing nearly 80% of the hospital care services for the whole territory, have adopted a population-based funding model in examining the relationship between the population and public health care. Population size and demographic structures are two core components of the population-based funding model to project territory-wide health care needs and to estimate the health care expenditures of Hong Kong in the short-to-medium term [17]. 

Analyses on the long-term effects of population ageing on health care needs have repeatedly emphasised that the increasing demand for health services due to the ageing population is a predictable, but inevitable, phenomenon [3]. In an earlier work [13], the authors compiled a 29-year projection on the hospitalisation needs of Hong Kong residents from 2000 to 2029 through a constant rate approach. They estimated that 80% of the projected increase in patient days would be contributed by the ageing effect, and by 2029, the older patients aged 65 years or above would account for more than 60% of the projected patient days. In a subsequent decomposition analysis [3], it was identified that the gradual reduction in the average LOS has contributed to a 60% reduction in the total patient days during the period 2001–2004. However, the magnitude of the LOS reduction was entirely offset by the combined effects attributable to population ageing, increase in the number of discharge episodes per patient, and the population growth from 2005 to 2012. Between 2012 and 2041, population ageing is projected to account for 60% of the projected LOS growth in Hong Kong. Furthermore, due to extra-low fertility, the population growth has reduced to about 0.5% and the population size will peak at 8.7 million and start to decrease from 2050 onwards [16]. These demographic changes would impose substantial impact on the hospitalisation needs of Hong Kong in the future. 

In the present analysis, the aim is to fill the gap by applying the constant rate approach in projecting the population’s inpatient needs for the next 55 years from 2011 to 2066 with the latest United Nation’s population projection [16]. Previous analyses have emphasised that population ageing is a prominent driving factor on the LOS increase, but none of them has attempted to examine whether owing to a demographic transition into a more stable distribution, in the longer term, the ageing effect would be slowed down. To better understand the potential impact of longevity prolongation on health care needs on population health aspects, in the latter part of the analysis, the present study also assesses the expected lifespan of staying in hospital and the expected hospitalisation-free time at birth, which are absolutely crucial for effective planning of hospital services in the community. 

Recent literature, see ref. [18] for example, showed that the effect of demographic change on the demand for health care services is highly overestimated if proximity to death is not controlled for. The study in ref. [18] was conducted using Hospital Episode Statistics, of English administrative data, of which a panel structure was in place to follow the individuals over seven years of this administrative data. Unfortunately, due to the limitation of the data, proximity to death cannot be taken into account in the present analyses, which is one of the limitations on the main findings of the present paper.

## 2. Data and Method

### 2.1. Data

To illustrate the factors contributing to the change in the days of hospitalisation in Hong Kong, population data and data on hospital discharges and admissions were collected from the Census and Statistics Department (C&SD) and the Hospital Authority (HA), respectively. All data collection and storage procedures were approved by the Human Research Ethic committee of The University of Hong Kong (ethical code: EA1707016). The population statistics are stratified by gender and five-year age groups with an open age group for 85 years or above [19]. To assist the calculations of the projections, life tables from 2000 to 2066 generated by the C&SD are also employed [20].

The HA’s microdata on the Hong Kong public hospital system provide detailed inpatient data of all public hospitals of Hong Kong for the period 2000–2010. For the purpose of the current numerical studies, aggregated information was generated to include the total LOS in hospital, the number of admissions and the number of inpatients by gender, age (0, 1, …, 94 and 95+), and the year (2000, …, 2010). 

### 2.2. Methodology

#### 2.2.1. Notations

i—Age-gender groups (gender: male and female; age group: individuals ageing from 0 to 94 and 95+; in total 192 groups)

$P_{it}$—Population size in group i in year *t*

$P_{t}$—Total population size in year *t*

*LOS_it_*—LOS in group *i* in year *t*

*TLOS_t_*—Total LOS in year *t*

$s_{t}\left(x\right)—$Survival probability of an individual born in year *t* to age *x*

$LTD_{t}—$Expected total days of hospitalisation before age 95 for one born in year *t*

*HFY_t_*—Expected total hospital-free time in the years before age 95 for one born in year *t*

#### 2.2.2. Mathematical Formulation–Projection

To assess the future impact of ageing on hospitalization in the population of Hong Kong, making use of the latest data from the C&SD, the total length of stay in hospital (TLOS) was projected up to 2066. In particular, it was assumed that the length of stay in hospital per capita for each age group *j* (0–4, 5–9, …, 80–84, 85+) would remain at the 2010 level for the next 56 years. For time *t*, *t* = 2011, …, 2066, then
(1)$$LOS_{jt}=LOS_{j,2010}\times \frac{P_{jt}}{P_{j,2010}}$$
and $TLOS_{t}=\sum_{j}LOS_{jt}$. Note that $LOS_{jt}$ and population size $P_{jt}$ in (1) were aggregated based on the single-age data, and the last age group 85+ was chosen according to the C&SD projection specification.

To better illustrate how the ageing of the Hong Kong population could increase the burden of the Hong Kong health system from 2010 to 2066, we estimated the total inpatient costs in Hong Kong public hospitals on a yearly basis. To simplify our calculations, we first assumed a constant general inflation rate of 3% per annum in Hong Kong over the whole period from 2010 to 2066. Then we predicted the average daily inpatient costs in Hong Kong public hospitals from 2019 to 2066 according to the 2010–2018 figures, which can be found in the HA annual reports and the Controlling Officer’s Reports in the government’s budget [21,22,23,24]. According to the 2010–2018 figures, the average annual medical inflation rate is around 5.2% per annum. To avoid over-discounting or over-accumulating, we used 5.2% to predict the next ten years’ daily inpatient costs, i.e., for the period 2019–2028. After 2028, we assumed the medical inflation rate would be equal to the general inflation rate. 

In addition, making use of the single-age data, a projection on $LTD_{t}$, the expected total number of days an individual will spend in hospitals, *HFY_t_*, the expected total hospital-free time, from birth in year *t* to age 95, and for *t* = 2010, …, 2066 was performed. This projection was conducted for females and males separately with the following assumptions:For each individual, his/her lifetime mortality experience is fully represented by the life table associated with his/her birth year, and the survival probabilities from birth $s_{t}\left(x\right)$ can be calculated accordingly. The life tables adopted are the Hong Kong Life Tables 2000–2066 produced by the C&SD of the Hong Kong SAR.Since detailed age structure of the population of Hong Kong for those aged 85 years and above is not available, to calculate the LOS per capita of the five-year age groups up to 95+, an estimation was made based on the 2000–2009 age structure data with the associated life tables. Assuming that the 84+ population in each year is a closed and stationary population, the people aged 84 in 2000 will turn to age 94 in 2010 (with probability *s*_2010_(94)/*s*_2010_(84)); the people aged 84 in 2001 will turn to age 93 in 2010 (with probability *s*_2011_(93)/*s*_2011_(84)), and so on. As a result, the age structure was extended from 0, …, 84, 85+ to 0, …, 94, 95+ and the genderwise LOS per capita was calculated using the aggregated data from the HA for 2010 for all singles ageing from 0 to 94.For individuals born after 2010, the average number of days in hospital for each future year of age is assumed to follow the 2010 LOS per capita figures, which was discussed in the above-mentioned bullet point.Due to lack of detailed data and the difficulty of making any reasonable assumptions, the age range 95+ was not considered.

For each gender, the formula for calculating $LTD_{t}$ is, for *t* = 2010, …, 2066,
(2)$$LTD_{t}=\overset{94}{\underset{x=0}{\sum }}\frac{LOS_{x,2010}}{P_{x,2010}}s_{t}\left(x+1\right)$$

Note that the single-age information up to age 95 available in the HA microdata and the single-age survival probabilities available in the Hong Kong life tables up to 2066 recently released by the C&SD were used. One advantage of incorporating single ages into the calculations rather than age groups is to avoid significant bias by making unrealistic assumptions. For example, when considering the probability of a new birth to survive to a given age range, one common assumption is to use the middle age in the range to represent the whole range. By so doing, the probabilities of the younger ages are underestimated and the older ages overestimated. This is also the main reason not to include the last age group 95+ into the calculation. 

## 3. Results

### Population Ageing Remains a Prominent Upward Force of Health Care Burden in Hong Kong

Figure 1 shows the projected TLOS in the future, assuming that since 2010, there is a stable pattern of hospitalisation. In the next 35–36 years, the TLOS numbers are expected to increase steadily. However, due to the stabilized older age group with the decreasing young and middle aged population, the speed of increase of the TLOS numbers starts to steadily reduce after 2038 with a trend of decrease from 2048. These observations are backed up by Figure 2 which contains the annual percentages of increase in the TLOS from 2010. For the period 2010–2066, the LOS in the age group 85+ is projected to increase by 555.3% while the LOS for the whole population is expected to increase by only 134.4%. In 2010, the proportion in the LOS contributed to by the oldest age group (85+) was 15%. In 2066, this proportion is projected to be nearly tripled (42%). Around 70% of the projected days of hospitalisation would be taken by people aged 75 years and above. It requires a significant shift in the whole health care system to address this matter. It is however worth noting that the above projection only reflects the sole contribution from the ageing effect of the population of Hong Kong on the total LOS in the period 2010–2066. 

Under the previous assumptions we made on how to estimate the total inpatient costs in Hong Kong public hospitals, we produced the following Table 1 that gives estimates of total hospitalisation costs in HKD (millions) for several specifically chosen age groups as well as on an overall scale from 2010 to 2066. For a fairer comparison, all values are discounted back to 2010 at the assumed constant general inflation rate of 3% per annum (except the Inpatient cost/day column).

The first observation from Table 1 is that the total inpatient hospitalisation costs nearly double from 2010 to 2036, reach a peak in 2051 and then decrease slightly after that. Secondly, all age groups show a strong increasing trend from 2010 to 2027, then the younger age groups start to go downwards. The older the age group, the later its turning point appears. The total inpatient hospitalisation expenses on people aged 85 and above keep increasing till 2056, which shows a nearly 30-year longer increasing period than people below age 70. Further, in 2010, the total cost by the 85+ age group is roughly 28.6% of the total cost by the age group 0–69, but in the year 2066, the percentage becomes 191% which is very strong evidence on how ageing could increase the total inpatient hospitalisation expenses under our assumptions. 

As illustrated in Figure 3, the projected increase in the TLOS for the older age groups would be more prominent and the peak would appear later. For example, it is projected that the LOS of the age group 65–74 would peak in the year 2031 while the corresponding figures for 75–79, 80–84, and 85+ would peak in 2041, 2046, and 2056, respectively. Moreover, Figure 4 shows the trend of the LOS in the five-year age groups (in total 17 such age groups plus the group of 85+) from 2000 to 2010. It agrees with the obvious shifting of the weights to the older ages, in particular, to the oldest ages. It is a good complement to Figure 1 in demonstrating the impact of ageing on the LOS. The stacked column chart, Figure 5, visualizes the percentage proportions of the 20 age groups from a different angle. Again, the trend of ageing is clearly shown from the top section of the graph. For the group 75+, its share in the total LOS increased from 29.7% in 2000 to 38.8% in 2010; the share of 85+ increased from 9.6% to 15%; the share of 95+ increased from 0.85% to 1.77%. The older the age, the higher is the increase.

Figure 6 presents the projected number of days of hospitalisation and hospital-free years at birth across birth cohorts. Two sets of results are projected for the years 2010 to 2066. Note that unless death has been observed before 2011, since the majority of the lifetime of the individuals born in the year is unknown, a projection is also necessary for the year 2000. The first set of results is illustrated by stacked columns with units of days. The columns show the expected total number of days a person spends in hospitals from birth to age 95 years for different birth years. It can be seen that starting from 2010, both genders have a general increasing trend. The number for females would increase from 108.5 days in 2010 to 139.3 days in 2066, while for males the number would increase from 106 days in 2010 to 143.8 days in 2066, displaying a faster speed of increase. Also, starting from 2031, it is projected that males would spend longer time in hospital during their lifetime than females, which could be interpreted as males in general falling ill more often than females, as the latter have longer life expectancies. Since the sole hospitalisation experience in 2010 is adopted for the whole projection period, the slow increasing trend for both females and males could be assigned to the expected improvement in mortality experience in the future.

The second set of results is demonstrated by two lines with units of years. For each year, the hospital-free time of an individual is the number of hospital-free days divided by the total number of days in the year. For both males and females, steady increasing trends are observed with minor fluctuations. Since the average LOS per person each year is relatively small, the increasing trends are largely contributed to by the general improving mortality experience of the people in Hong Kong, i.e., the effect of ageing. 

## 4. Limitations

The present analysis adopted a fixed rate approach, which assumes the LOS is only driven by demographic factors, and any improvements on health technologies and health awareness have not been accounted for. Only the overall inpatient data from the HA were used. According to the categorisation of health care services at the HA, inpatient care is subgrouped into acute inpatient (medical, surgical, intensive care), non-acute inpatient (mostly rehabilitative/mental health), and day hospital (chemotherapy, peritoneal dialysis). Costs and characteristics of these cares are different and may not be fully comparable. To reduce the costs and promote community care, the HA has devoted efforts to reduce the episodes and average the LOS of non-acute inpatient care (that is why an overall LOS reduction during the early 2000s was observed). On the other hand, acute inpatient and day hospital are unlikely to be avoidable. Further, the latest population projections from the United Nations and the C&SD are used, but they are also subject to change. Thus, more continuous monitoring and surveillance are needed.

The numerical values in Table 1 were estimated based on the simple general inflation rate assumption and our proposed method of predicting daily inpatient costs in Hong Kong public hospitals for the period of 2019–2066. These assumptions are far from perfect as they do not consider a lot of factors like age effect, health technology advances, cost differences among major service types of inpatient care and speciality. The only purpose in producing Table 1 is to showcase the impact of population ageing on the Hong Kong public health system in monetary form.

## 5. Conclusion

Hospitalisation remains most prominent in compiling the health care expenditure of any health system in the world. Increase of the elderly population would unavoidably increase the burden of the hospital system. The change in the age-specific population structure has indeed shifted hospital care towards older adults. The findings of this study suggest that the proportion of hospital care for older adults has increased from 17% to about 43%. Thus, the health care system should respond to this challenge by providing sufficient training for geriatric health care services in Hong Kong. More effort on strengthening primary health services should be focused to reduce ambulatory care-sensitive hospitalization, otherwise, the existing system will simply not be able to cope with the ever increasing needs. Further, more innovative measures such as a community-based care system should be explored to provide support for the overstretched hospital system. 

## 6. Discussion

There are several semi-private hospitals which have been set up in the last few years in Hong Kong to help mitigate the burden of the public hospital system. As with the opening of the Gleneagles Hong Kong Hospital of the University of Hong Kong, the Chinese University of Hong Kong Medical Centre and other private hospitals, patients with a more affluent background should be diverted to the private sector. At the same time, the Hong Kong SAR Government is promoting a type of universal insurance coverage for the general population, however, its potential impact has yet to materialize.

The population growth in Hong Kong has slowed down in recent years with less than 0.5% growth annually. It is projected that the trend will continue until the 2050s. By then, the population mix would be more balanced and the hospital needs might be different. However, before 2050, there is still a great need to provide sufficient hospitalisation services over the next two decades in Hong Kong. 

The government expenditure on health care in Hong Kong has been increasing at an alarming rate at more than 10%–15% per annum, and much effort has been spent to identify more cost-effective and better-quality hospital services in the community. Certain medical procedures like peritoneal dialysis and chemotherapy can be performed at home or in day units, thus the patients are no longer required to be admitted. Nonetheless, if these medical procedures are to be performed outside a hospital setting, relevant measures and facilities should be taken to protect the patients from infections or other adverse health outcomes. How related technology advancement and patient management can help to reduce the days of hospitalisation is absolutely crucial to contain expenditure. It is thus important to have the empirical figures available to objectively assess the needs of hospitalization in the future. 

## Figures and Tables

**Figure 1 ijerph-16-00473-f001:**
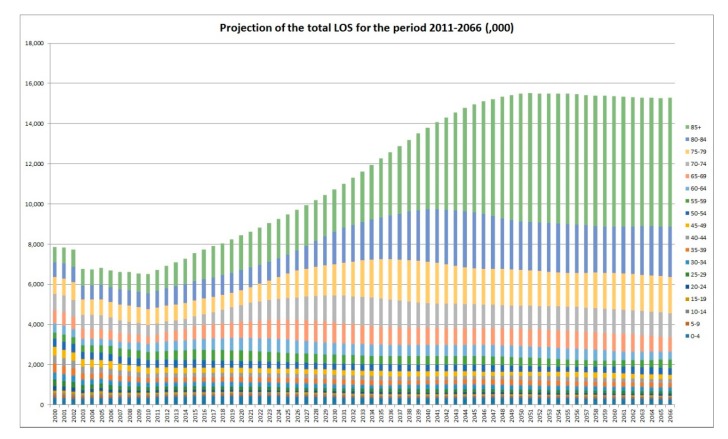
Total days of hospitalisation: 2000–2066. LOS: length of stay.

**Figure 2 ijerph-16-00473-f002:**
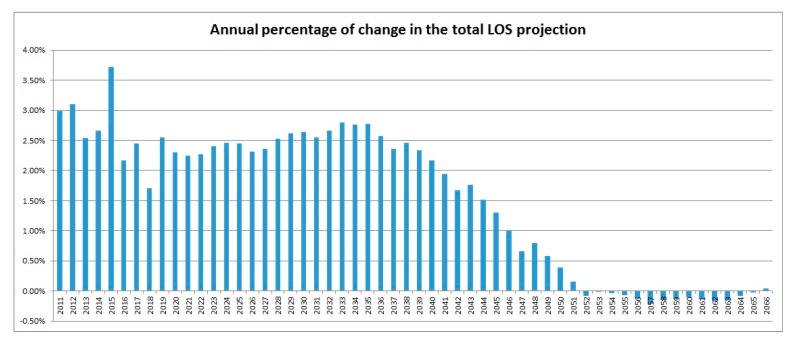
Annual % of changes in the TLOS: 2011–2066. TLOS: the total length of stay.

**Figure 3 ijerph-16-00473-f003:**
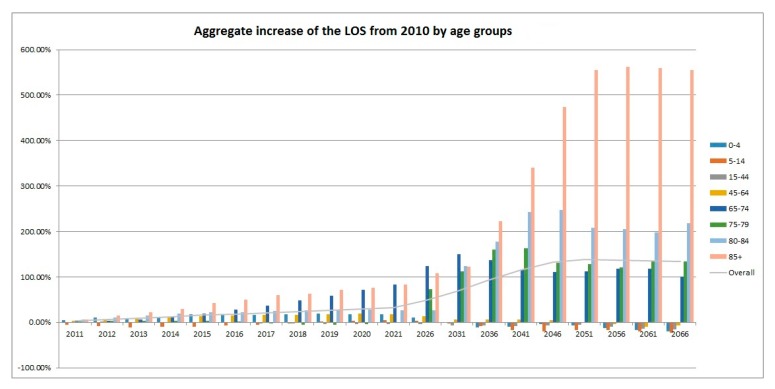
Aggregate increase in the LOS from 2010: 2011–2066.

**Figure 4 ijerph-16-00473-f004:**
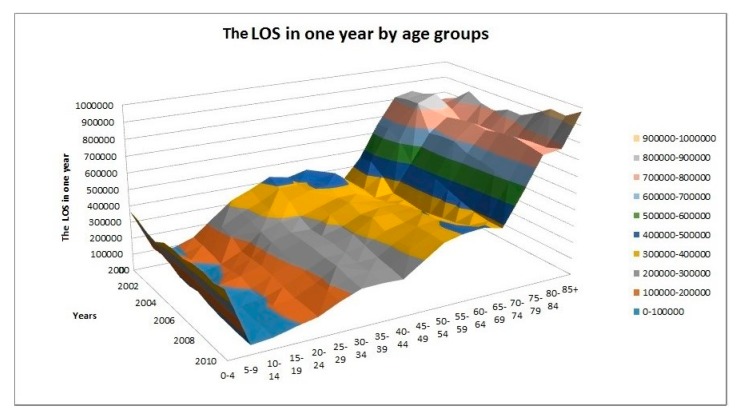
Total length of stay in hospital in each year by age group.

**Figure 5 ijerph-16-00473-f005:**
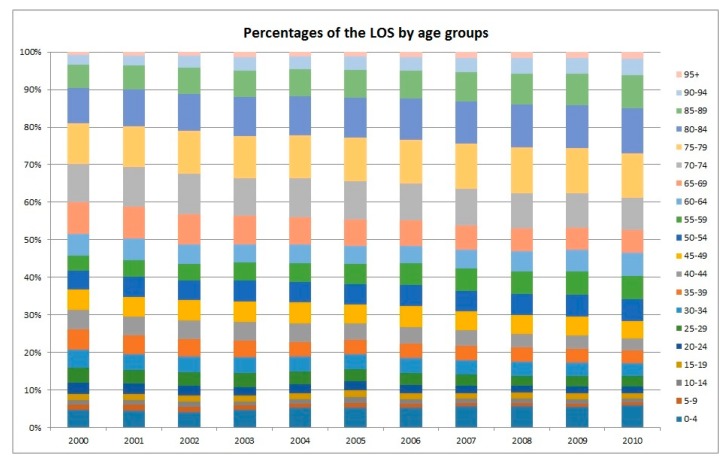
Percentages of the LOS by age group column chart.

**Figure 6 ijerph-16-00473-f006:**
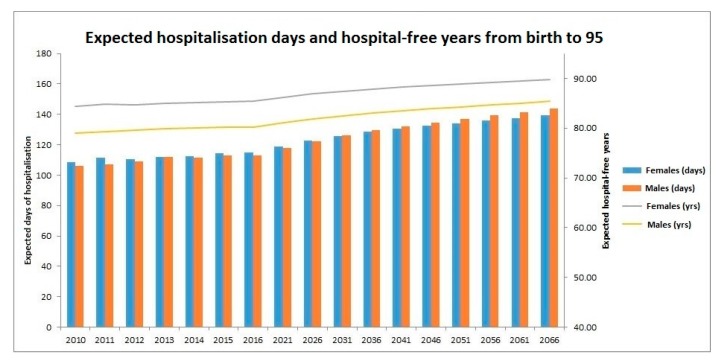
Projected days of hospitalisation and hospital-free years at birth by cohort.

**Table 1 ijerph-16-00473-t001:** Total discounted hospitalisation costs by age groups 2010–2066 (HKD Million).

Year	Inpatient Cost/Day	0–69	70–74	75–79	80–84	85+	Total
2010	HKD 3600	12,317.9	2041.6	2763.7	2816.1	3526.1	20,858.2
2011	HKD 3950	13,426.4	2145.0	2968.7	3194.5	4008.7	21,481.1
2012	HKD 4180	14,240.6	2114.9	3105.1	3392.2	4417.5	22,148.1
2013	HKD 4330	14,615.4	2050.0	3142.2	3560.7	4753.2	22,709.8
2014	HKD 4600	15,436.7	2098.5	3227.1	3838.0	5178.5	23,315.6
2015	HKD 4830	16,211.9	2168.6	3308.6	3986.1	5810.4	24,182.5
2016	HKD 4950	16,442.2	2234.2	3240.4	3985.1	6106.2	24,707.4
2017	HKD 5270	17,167.5	2593.4	3213.6	4216.8	6704.5	25,313.1
2018	HKD 5390	17,257.2	2910.9	3078.7	4211.5	6774.5	25,745.5
2019	HKD 5670	17,829.7	3288.5	3139.6	4313.8	7285.0	26,402.9
2020	HKD 5965	18,344.0	3750.9	3255.2	4446.4	7670.0	27,011.6
2021	HKD 6275	18,879.9	4176.6	3458.1	4500.0	8113.4	27,619.6
2022	HKD 6601	19,411.0	4441.1	4021.0	4411.3	8587.7	28,247.6
2023	HKD 6945	19,961.0	4703.6	4627.8	4381.5	9074.6	28,926.7
2024	HKD 7306	20,447.0	5013.9	5245.9	4494.3	9534.1	29,638.2
2025	HKD 7686	20,903.6	5301.8	5994.4	4683.1	9926.8	30,364.3
2026	HKD 8085	21,298.7	5679.2	6677.4	5004.3	10,256.4	31,067.2
2027	HKD 8506	21,614.5	6120.3	7115.1	5847.6	10,441.1	31,799.8
2028	HKD 8761	21,489.0	6455.1	7386.9	6613.4	10,488.3	32,604.5
2029	HKD 9024	21,327.7	6706.2	7718.9	7364.6	10,685.6	33,456.5
2030	HKD 9294	21,125.1	6934.9	8004.2	8250.6	10,908.7	34,339.9
2031	HKD 9573	20,896.1	7091.0	8404.2	8995.9	11,243.3	35,214.7
2036	HKD 11,098	19,894.6	7116.0	10,304.8	11,187.7	16,232.2	40,254.6
2041	HKD 12,866	19,753.3	6211.5	10,394.1	13,784.3	22,220.6	44,998.3
2046	HKD 14,915	19,645.7	6107.8	9126.4	14,004.4	28,895.3	48,366.0
2051	HKD 17,290	19,476.2	5904.2	9004.1	12,408.2	33,026.3	49,634.2
2056	HKD 20,044	18,855.9	6249.0	8740.0	12,299.7	33,412.4	49,471.2
2061	HKD 23,237	18,186.7	6219.0	9283.6	12,009.2	33,258.0	49,097.8
2066	HKD 26,938	17,288.7	6229.0	9270.0	12,819.0	33,030.6	48,899.4

In column 2, the 2010–2018 figures are obtained from the Hong Kong Hospital Authority and Hong Kong government reports. The 2019–2028 figures are projections based on 5.2% increase per annum using the 2018 figure, and the 2029 onwards figures are projections based on 3% increase per annum using the 2028 one. The values in columns 3–8 are discounted values at 2010 at the rate 3% per annum.

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
