# Peer review of "A Projection of Future Hospitalisation Needs in a Rapidly Ageing Society: A Hong Kong Experience"

_ijerph, 2019, doi:10.3390/ijerph16030473_

Round 1

Reviewer 1 Report

Dear Author,

in my opinion your work is interesting and the survey used is very rich of information.

However, I think that your results could be more helpful for the health national system if you could add an estimate of costs. How much expensive will be the national health system if the population will increase as your suggestion?

In this form, in my opinion is too simple. You should evaluate also the financial costs considering a mean cost per day of hospitalization for the entire period that you have taken into consideration.

Author Response

Dear Reviewer,

Greetings

thank you suggestion, we have included an expenditure information in Table 1

the cost of a day in the hospital  is HKD5390   in 2018 which is   based on the latest hospital figure.

We have also made the projection up to 2066 as given in Table 1. The 2019-2028 figures are projections based on 5.2% increase per annum using the 2018 figure, and the 2029 onwards figures are projections based on 3% increase per annum using the 2028 one.

The values in columns 3-8 are discounted values at 2010 at the rate 3% per annum.

indeed it has highlighted the financial burden of the ageing. 

thank you for the suggestion 

Cheers
Paul

Reviewer 2 Report

The article is very well-written and presented, despite you have missed to present results in Tables at the end and to justify better how you can't deepen the Major Diagnostic Categories (MDC) analysis. Indeed it might constitute a major novelty to concentrate also you model on some precise expenditures flows, if possible.

Author Response

thank you for the suggestion.

the paper is mainly to highlight the ageing impact on the hospitalization days.

the suggestion would be relevant if we are interested in some major Diagnostic Categories 

which will change the focus of the paper and substantially increase the length of the paper. 

we shall certainly consider this in the following up paper. 

Thank you for your attention 

Cheers
Paul

Round 2

Reviewer 1 Report

Dear Author,

I think that now your work is more interesting and usefull also for policy makers.

All the best